# Is SARS-CoV-2 Neutralized More Effectively by IgM and IgA than IgG Having the Same Fab Region?

**DOI:** 10.3390/pathogens10060751

**Published:** 2021-06-13

**Authors:** Yalcin Pisil, Zafer Yazici, Hisatoshi Shida, Tomoyuki Miura

**Affiliations:** 1Laboratory of Primate Model, Research Center for Infectious Diseases, Institute for Frontier Life and Medical Science, Kyoto University, Kyoto 615-8530, Japan; yalcin.pisil@gmail.com; 2Department of Virology, Faculty of Veterinary Medicine, 19 Mayis University, Samsun 55270, Turkey; zyazici@omu.edu.tr; 3Division of Molecular Virology, Institute of Immunological Science, Hokkaido University, Hokkaido 060-0808, Japan; hmyy2010@yahoo.co.jp

**Keywords:** SARS-CoV-2, COVID-19, neutralizing antibody, IgG, IgM, IgA

## Abstract

Recently, recombinant monoclonal antibodies (mAbs) of three Ig isotypes (IgG, IgA, and IgM) sharing the same anti-spike protein Fab region were developed; we evaluated their neutralizing abilities using a pseudo-typed lentivirus coated with the SARS-CoV-2 spike protein and ACE2-transfected Crandell–Rees feline kidney cells as the host cell line. Although each of the anti-SARS-CoV-2 mAbs was able to neutralize the spike-coated lentiviruses, IgM and IgA neutralized the viral particles at 225-fold and 125-fold lower concentrations, respectively, than that of IgG. Our finding that the neutralization ability of Igs with the same Fab domain was dramatically higher for IgM and IgA than IgG mAbs suggests a strategy for developing effective and affordable antibody therapies for COVID-19. The efficient neutralization conferred by IgM and IgA mAbs can be explained by their capacity to bind multiple virions. While several IgG mAbs have been approved as therapeutics by the FDA, there are currently no IgM or IgA mAbs available. We suggest that mAbs with multiple antigen-binding sites such as IgM and IgA could be developed as the new generation of therapy.

Severe acute respiratory syndrome-coronavirus-2 (SARS-CoV-2) has caused the second pandemic of the 21st century, which so far has killed more than 2.5 million people and infected more than 130 million in 223 territories globally (World Health Organization, 2021).

Current clinical management procedures for coronavirus disease 2019 (COVID-19) include good hygiene practice, infection diagnosis, and supportive care such as supplemental oxygen and mechanical ventilatory support. The United States Food and Drug Administration (FDA) has approved one drug, remdesivir (Veklury), for the treatment of COVID-19; however, the World Health Organization does not currently recommend usage of remdesivir. Favipiravir, an approved drug to treat influenza, has also been administered to treat COVID-19, but its antiviral efficacy is still under debate [1]. Unfortunately, no effective antiviral treatments for COVID-19 are currently available due to our limited knowledge about SARS-CoV-2 and lengthy drug development time frames [2].

Antibodies collected from convalescent individuals can be used to treat infectious diseases. Approximately 90% of individuals with mild-to-moderate COVID-19 produce anti-SARS-CoV-2 antibodies. Immunoglobin (Ig) M and IgA are typically produced within 7 days [3], and IgG development occurs 10–18 days post-infection; antibody titres remain stable for at least 5 months after infection [4]. Convalescent plasma containing neutralizing antibodies may be able to modulate the inflammatory response of newly infected COVID-19 patients and could therefore be used as a therapy for COVID-19 [5]. However, antibody therapies carry the risk of triggering allergic/anaphylactic reactions, white blood cell/red blood cell alloimmunization, lung damage and difficulty breathing, haemolytic transfusion reactions, and infections [6]. Moreover, the levels of virus-neutralizing antibodies in convalescent plasma are often too low for effective treatment [7].

The passive administration of monoclonal antibodies (mAbs) is a promising antiviral therapy for AIDS and COVID-19 [7,8,9]. Several anti-SARS-CoV-2 mAbs have been isolated from the B cells of infected individuals. The majority of the mAbs isolated target the receptor-binding domain of the SARS-CoV-2 spike protein, which interacts with the angiotensin-converting enzyme 2 (ACE2) receptor to initiate the infection process [10,11,12]. The isolated mAbs effectively neutralize SARS-CoV-2 in vivo [9,13,14,15]. Combining multiple mAb clones can have a synergetic effect on neutralizing SARS-CoV-2 by recognizing different epitopes of the receptor-binding domain. Combination treatment of the anti-SARS-CoV-2 mAbs casirivimab and imdevimab has been approved by the FDA for use in mild-to-moderately ill high-risk patients [16].

The mass production of mAbs is laborious and expensive. Thus, researchers are searching for ways to increase mAb potency and reduce the concentration of mAbs required for effective treatment. The five classes of Igs are IgM, IgD, IgG, IgA, and IgE. All Ig molecules contain a fragment antigen-binding (Fab) region, which recognizes antigens, and fragment crystallizable (Fc) regions, which mediate the effector functions of natural killer cells, macrophages and the complement system. IgG is monomeric, IgM is multimeric (typically pentameric), and IgA exists in both monomeric and dimeric forms. The number of antigen molecules trapped by each Ig molecule can influence the effectiveness of virus neutralization [17].

Recently, recombinant mAbs of three Ig isotypes (IgG, IgA, and IgM) sharing the same anti-spike protein Fab region were developed; we evaluated their neutralizing abilities using a pseudo-typed lentivirus coated with the SARS-CoV-2 spike protein and ACE2-transfected Crandell–Rees feline kidney cells as the host cell line [18].

Although each of the anti-SARS-CoV-2 mAbs was able to neutralize the spike-coated lentiviruses, IgM and IgA neutralized the viral particles at 225-fold and 125-fold lower concentrations, respectively, than that of IgG [18] (Figure 1a). Our finding that the neutralization ability of Igs with the same Fab domain was dramatically higher for IgM and IgA than IgG mAbs suggests a strategy for developing effective and affordable antibody therapies for COVID-19. The efficient neutralization conferred by IgM and IgA mAbs can be explained by their capacity to bind multiple virions [18] (Figure 1b). Underlying reasons could be the enrichment of cross-linking of viral antigens, complement fixing, and the neutralization of virus-infected cells. This is consistent with a recent report in which the monomeric form of anti-SARS-CoV-2 IgA found in serum was two-fold less potent than IgG, while the dimeric, secretory form of IgA was 10-fold more potent than monomeric IgA [19].

Only the IgG type of mAbs has been clinically applied so far. The reasons could be ascribed to difficulty of IgM purification compared to IgG and the less-stable nature of IgM [20]. But these factors can be overcome by the recent progress of techniques such as new chromatography strategies and Fc glycan modifications [21,22]. Additionally, dimeric IgA and polymeric IgM can bind polymeric immunoglobulin receptors (pIgR). Owing to pIgR, dimeric IgA and polymeric IgM are transferred from the lamina propria across the epithelial barrier to mucosal surfaces [23]. Therefore, IgA and IgM could be injected intravenously and also administered via the nasal pathway, delivering to mucosal organs including the lungs.

While several IgG mAbs have been approved as therapeutics by the FDA, there are currently no IgM or IgA mAbs available. Our finding about efficacy of the polymerization of antibodies suggests to pharmaceutical companies that mAbs with multiple antigen-binding sites such as IgM and IgA could be developed as the new generation of therapy.

## Figures and Tables

**Figure 1 pathogens-10-00751-f001:**
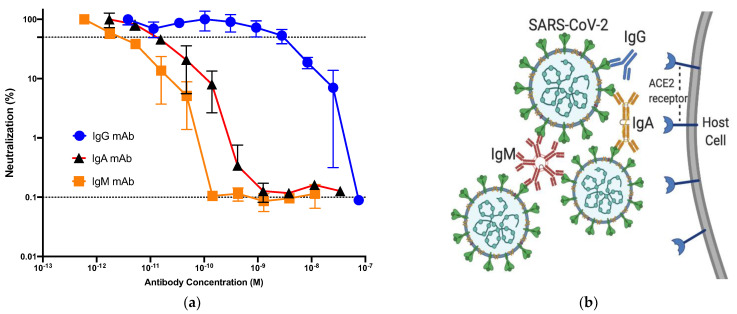
**(a)** Neutralization of pseudo-typed lentivirus coated with the SARS-CoV-2 Spike protein (LpVspike(+)) by anti-SARS-CoV-2 monoclonal antibodies (mAbs). After pre-incubating LpVspike(+) with each anti-SARS-CoV-2 neutralizing mAb at a 100 TCID50 (50% tissue culture infectious dose), the mAb/virus mixtures were added to ACE2-expressing CRFK cells and cultured for 48 h, after which luciferase activity was measured. The IgG, IgM, and IgA mAbs were diluted serially three-fold, from an initial concentration of 10 μg mL^−1^ to 0.016 μg mL^−1^. The x- and y-axes are depicted in logarithmic scale [18]. (**b**) Neutralization of SARS-CoV-2 by three anti-SARS-CoV-2 neutralizing mAbs (IgG, IgM, and IgA). IgG has two antigen-binding sites, while dimeric IgA has four antigen-binding sites. Pentameric IgM has 10 antigen-binding sites and can bind 10 small antigens; however, due to steric restrictions, only five large viral antigens can be bound by one IgM molecule. IgG can bind to only one large antigen, whereas dimeric, trimeric, and pentameric IgA can bind to multiple large antigens.

## Data Availability

The data that support the findings of this study are openly available in https://doi.org/10.3390/pathogens10020153.

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
