# Peer review of "Is SARS-CoV-2 Neutralized More Effectively by IgM and IgA than IgG Having the Same Fab Region?"

_pathogens, 2021, doi:10.3390/pathogens10060751_

Round 1

Reviewer 1 Report

This is useful idea and helpful review of the antibody-mediated therapy for SARS-CoV 2 infections. The importance of valency of antibodies is well known phenomenon.

This should be described more details with respect to the underlying reasons such as cross-linking of viral antigens, complement fixing and neutralization of virus-infected cells.

The overall improvement of efficacy by polymerization of antibodies can be emphasized.

Author Response

To whom it may concern

Thank you a lot for your very kindly feedback and comment.

We added your recommendation end of the paper were highlighted in yellow.

The sentence where highlighted by green was changed for better English grammarly.

Sincerely

Reviewer 2 Report

In this paper the Authors aim to evaluate the IgM and IgA neutralizing activity to SARS-CoV-2 compared to IgG. A comprehensive and extensive literature review of the NCBI database PubMed was also carried out. The article was well conducted and it is interesting in its fields. It is a well-structured paper, written in good English and the References are up dated. 

Minor issues:

The during COVID-19 pandemic, the clinical evaluation of patients is deeply changed. The use of telemedicine and remote counselling, in fact, has gained great importance during covid 19 pandemic also in surgical fileds. In the “discussion” section I suggest to better analyze this topic. Therefore, the following paper should be considered:

“Gambardella C, Pagliuca R, Pomilla G, Gambardella A. COVID-19 risk contagion: Organization and procedures in a South Italy geriatric oncology ward. J Geriatr Oncol. 2020 May 22:S1879-4068(20)30237-X. doi: 10.1016/j.jgo.2020.05.008.”

“Tolone S, Gambardella C, Brusciano L, Del Genio G, Lucido FS, Docimo L. Telephonic triage before surgical ward admission and telemedicine during COVID-19 outbreak in Italy. Effective and easy procedures to reduce in-hospital positivity. Int J Surg. 2020;78:123-125. doi:10.1016/j.ijsu.2020.04.060”

Author Response

To whom it may concern

Thank you a lot for your very kindly feedbacks and comment.

Unfortunately, our paper is unrelated to "telemedicine and remote counselling."

This topic could be assessable under another paper in future.

The green-highlighted sentence in the paper was changed for better English grammar.

However, we added some sentences where yellow-highlighted in the paper by other reviewer recommendations.

Sincerely